# Oleogel Systems for Chocolate Production: A Systematic Review

**DOI:** 10.3390/gels10090561

**Published:** 2024-08-29

**Authors:** Jheniffer E. Valdivia-Culqui, Jorge L. Maicelo-Quintana, Ilse S. Cayo-Colca, Marleni Medina-Mendoza, Efraín M. Castro-Alayo, César R. Balcázar-Zumaeta

**Affiliations:** 1Instituto de Investigación, Innovación y Desarrollo para el Sector Agrario y Agroindustrial (IIDAA), Facultad de Ingeniería y Ciencias Agrarias, Universidad Nacional Toribio Rodríguez de Mendoza de Amazonas, Chachapoyas 01000, Peru; 7572976720@untrm.edu.pe (J.E.V.-C.); marleni.medina@untrm.edu.pe (M.M.-M.); efrain.castro@untrm.edu.pe (E.M.C.-A.); 2Facultad de Ingeniería Zootecnista, Agronegocios y Biotecnología, Universidad Nacional Toribio Rodríguez de Mendoza de Amazonas, Chachapoyas 01000, Peru; jmaicelo@untrm.edu.pe (J.L.M.-Q.); icayo.fizab@untrm.edu.pe (I.S.C.-C.); 3Programa de Doctorado en Ciencias Agrarias, Escuela de Posgrado, Universidad Nacional de Piura, Jr. Tacna 748, Piura 20002, Peru

**Keywords:** saturated fatty acids, cocoa butter, pectin

## Abstract

In response to the growing demand for healthier food options, this review explores advances in oleogel systems as an innovative solution to reduce saturated fats in chocolates. Although appreciated for its flavor and texture, chocolate is high in calories, mainly due to cocoa butter (CB), which is rich in saturated fats. Oleogels, three-dimensional structures formed by structuring agents in edible oils, stand out in terms of mimicking saturated fats’ physical and sensory properties without compromising the quality of chocolate. This study reviews how oleogels could improve chocolate’s stability and sensory quality, exploring the potential of pectin-rich agro-industrial by-products as sustainable alternatives. It also explores the need for physicochemical evaluations of both oleogel and oleogel-based chocolate.

## 1. Introduction

Growing health concerns have driven a global trend towards consumers seeking products that satisfy nutritional needs and offer additional health benefits [1]. In this context, foods such as chocolates are appreciated on the market not just for their nutritional value but also for the sensory experience they provide. The taste, aroma, and texture of chocolate, along with its characteristic slow melting in the mouth, offer a satisfying experience, gaining popularity worldwide [2,3].

The composition of chocolate typically includes cocoa butter (CB) as the continuous phase and sugar, cocoa liquor, and other ingredients as the dispersed phase [4,5]. Cocoa butter, which is a vegetable fat, plays a crucial role in determining the desirable properties of chocolate, such as its microstructure, texture, crystallization, rheological, and sensory characteristics [5,6,7]. These properties are largely influenced by the presence of specific triacylglycerols (TAGs) like 1,3-dipalmitoyl-2-oleoyl-glycerol (POP), 1-palmitoyl-2-oleoyl-3-stearoyl-glycerol (POS), and 1,3-distearoyl-2-oleyl-glycerol (SOS) [8,9]. These TAGs, combined with the fatty acids in cocoa butter, contribute to the creation of a stable colloidal network in chocolate [10,11,12]. However, the presence of saturated fatty acids, mainly in CB (60%) [4,13], causes chocolate to have a high-calorie content of approximately 500 kcal/100 g [14]. This is associated with an increase in blood cholesterol, increasing the risk of cardiovascular disease, obesity, and possibly cancer [15,16]. Against this backdrop, reformulating ingredients in chocolates is an option that aims to provide benefits beyond simple palate satisfaction by integrating ingredients that could improve health [17,18].

One promising way to mitigate the potential adverse effects of chocolate consumption is to reduce saturated fats through the use of alternative ingredients, thereby eliminating partially fatty acids derived from the CB [19,20], with a threshold on the intake of saturated fat, typically < 10% of energy per day according to World Health Organization (WHO) [21]. This not only contributes to chocolate’s stability and texture [14,22,23] but also opens up a world of possibilities for improving health outcomes. A particularly promising alternative is the use of oleogels, which, as previous studies have shown, can reduce fat content without compromising food quality [13,24,25,26]. Furthermore, oleogels have even been successful in delaying fat bloom in chocolate [27], offering a ray of hope for healthier chocolate consumption and the potential for a longer shelf life.

Oleogels, also known as structured oils or organogels [28], are a product of the alternative structuring process of edible oils that has piqued the interest of the scientific community [29,30]. The process of obtaining oleogels is a fascinating one, with Gandolfo et al. [31] being the first to report the structuring of oils using mixtures of fatty acids and fatty alcohols, laying the foundations for subsequent research. Oleogels are obtained via two methods: the direct method, in which structuring agents are dispersed directly in edible oils by shearing and heating (above their melting temperature) followed by cooling to form the oleogel [32], and the indirect method, which involves the creation of templates such as emulsions, hydrogels, foams, or solvent exchange [33]. This results in a crystalline structure of saturated fats that provide physical and textural properties similar to those of CB in chocolates [34]. This scientific process is not only intriguing but also holds the potential to revolutionize the chocolate industry.

It is essential to pay attention in the use of oleogels to the type of structuring agent [35], which can be crystalline particles (i), self-assembled structures of low-molecular-weight compounds (ii), self-assembled structures of polymers or polymer chains (iii) and various structures such as colloidal particles and emulsion droplets (iv) [30]; which, accompanied by an oil phase under suitable environmental and processing conditions can produce oleogels suitable for use in the chocolate industry [35,36]. In contrast, a limitation in the development of oleogels as fat substitutes includes the scarcity of structuring agents suitable for human consumption [37], creating the need to investigate novel ingredients such as waxes, monoglycerides, phytosterols, lecithin, cellulose derivatives, and pectin, among others [6,13,37,38,39].

As Lu et al. and Yaqoob et al. [40,41] suggest, pectin, an agro-industrial by-product extracted from certain fruits, such as apples and citrus, is a promising candidate structuring agent for oleogelation because it significantly improves the oxidative stability of the oleogel; however, it is barely used in the formulation of chocolates. This underutilized potential of pectins as a possible structuring agent for oleogels is an intriguing prospect for the chocolate industry.

In this study, a systematic search of the scientific literature of the last eight years was performed following the methodology of Maicelo-Quintana et al. [42], focusing on three points: edible oleogels, reducing saturated fat in chocolates, and preserving the desirable properties of chocolate. The review includes studies focused on the quality of oleogel-based chocolates, human health, oleogel composition, measurement of physicochemical properties of oleogels, and the identification of structuring agents in the chocolate industry. Key studies from previous years have also been integrated to provide a solid theoretical basis, enriching the analysis and allowing a deeper understanding of recent advances in this emerging field.

Therefore, this review aims to provide a comprehensive and critical evaluation of edible oleogels in chocolate formulations, highlighting the potential of pectin as a structuring agent for producing an oleogel. At the same time, the principles of oleogel formulation, its effect on chocolate’s sensory and functional properties, and possible risks to human health are addressed. By reviewing recent advances in the application of oleogels in food, we integrated these findings with previous theories and studies to provide a comprehensive view of how they have transformed the stability and quality of chocolates. This transformative impact of oleogels inspires optimism about the future of chocolate formulations, promoting healthier and more sustainable alternatives.

## 2. Edible Oleogels

Using edible unsaturated oils has led to the establishment of a type of oleogels known as organogels [43,44], which are complex fat-mimicking microstructured systems that include an edible oil phase trapped by a thermo-reversible three-dimensional gel network created by structuring agents that transform the mixture into a semi-solid structure [25,45,46]. In this process, non-covalent interactions of the agent with the oil occur, forming fibrillar or platelet crystals [46] through hydrogen bonds, π-π stacking, electrostatic interactions, and van der Waals forces [47,48]. According to Silva and Danthine [49], oleogels possess physical properties similar to solid fat but without a high content of saturated fatty acids [50]. This effectiveness has been demonstrated in numerous studies, which have consistently shown that oleogels are a reliable alternative for reducing saturated fats in food products without significantly compromising their quality [51,52,53,54,55], following the FDA regulation for the drafted and enacted trans fats elimination policies [21].

### 2.1. Edible Oleogel Elaboration

Oleogels are the result of an oleogelation process [49,56], where their properties, such as viscosity, texture, hardness, and melting point, depend on the type of structuring agent used [35,57], the liquid phase, environmental, and processing conditions [36,50]. Therefore, understanding the mechanism of oleogelation is essential. Furthermore, oleogelation traditionally occurs through the self-assembly of a structuring agent that is responsible for inducing viscosity and solid properties in oil-based systems [58]. Figure 1 and Figure 2 describe the process of making edible oleogels, which can vary according to the properties and proportions of the microstructured system (oleogel), preparation temperature and time, mixing method [45], and the different types of chocolates where they will be added.

Figure 1 shows that the sequence for oleogel formation starts with carefully selecting the structuring agent and the oil phase [59,60,61]. Subsequently, both are mixed and fused with thermal action according to the melting point of the structuring agent [62,63]. The stability will depend on the strength and nature of the molecular interactions between the structuring agent and the oil (Figure 2) [64]. After cooling, the oleogel is obtained [25,30]. The final product is suggested to be stored at low temperatures.

### 2.2. Technical Parameters Required for Edible Oleogel Elaboration

The direct method of producing oleogels, which uses high temperatures, makes it possible to disperse the structuring agents in edible oils (oil–liquid phase) at these temperatures (melting) depending on the oil phase [65], as stated in the scientific literature. For example, Trujillo-Ramirez et al. [66] increased the heating temperature of chia seed oil to 80 °C to effectively disperse glyceryl monostearate [67] and sorbitan monostearate [68], with a melting temperature range between 56 and 62 °C. Similarly, to making oleogels from ethylcellulose, the heating temperature of the oil must be higher than 100 °C [59,69,70], and when β-sitosterol and lecithin are used, the heating temperature of the oil is around 90 °C [34].

Understanding the melting temperature of each structuring agent is critical for obtaining a uniform mixture of oleogels as it determines the degree of oxidation of the oleogel and, consequently, promotes fat bloom in chocolate [26,71]. Further, achieving a homogeneous mixture requires constant mixing. For instance, in the preparation of oleogels of girasol oil with sodium stearoyl lactylate, the mixture was stirred for 30 min at 75 °C [72], while in a study conducted by Chen et al. [73], the mixture of a peanut diacylglycerol oil oleogel with ethyl cellulose and monoglycerides was stirred for 2 h at 120 °C. Similar variations in mixing time exist for different structuring agents [52,74], and more research is needed to determine the optimal dispersion time for each structuring agent as a substitute for CB in chocolate. Another factor to consider is the temperature during cooling and storage; this is because the cooling process allows for the formation of a three-dimensional network of the structuring agent, immobilizing the oil [43]. The cooling temperature is reported to vary between 24 and 48 h to achieve oleogel stability [75,76,77,78]. The storage temperature is suggested to be 4 °C [79,80,81]. However, it is necessary to identify the optimal cooling conditions according to the type of oleogel produced and the storage temperature to preserve the stability, texture, and viscosity of the oleogel.

In the case of the indirect method, the preparation of oleogels is based on the use of polysaccharides as structuring agents since they cannot interact directly with the oil [82] due to their hydrophilic and water-soluble properties in polar solvents (except for ethyl cellulose and chitin, which are also suitable for direct dispersion) [33]. One of the ways to obtain oleogels via this method is through the use of emulsion templating, which involves the initial formation of an emulsion followed by the removal of water to obtain the oleogel [83]. Another method is foam templating, where the hydrocolloid is hydrated in water to develop its structure, then freeze-dried and incorporated with the oil to form the oleogel [84]. Some of the structuring agents that form oleogels via this method are hydroxypropyl methylcellulose, methylcellulose, and zein [13,65]. For this method, it is also necessary to identify optimal storage conditions to maintain the quality.

### 2.3. Edible Oleogels Composition

Table 1 shows findings on the composition of edible oleogels in different foods, including chocolate. The choice of oils (oil phase) and structuring agents is vital to avoid negative effects on chocolate’s desirable properties [38].

#### 2.3.1. Oil Phase

The oil phase is not just a component but a crucial element of oleogels due to the fact that the fatty acids derived from vegetable oils are generally unsaturated [25]. Table 1 shows oil phases, such as sunflower oil (monounsaturated and polyunsaturated fatty acids) [141], safflower oil (linoleic acid of the omega-6 family) [142], soybean oil (low in saturated fats and cholesterol) [143], linseed oil (high percentage of unsaturated fats) [144], olive oil (high degree of unsaturation) [145], canola oil (a high content of monounsaturated fatty acids) [146], and palm and coconut oils (saturated fatty acids) [147]. In chocolate, substituting CB with vegetable oil-based oleogels could affect its organoleptic characteristics, given the chemical nature of the substituted fatty acids [148,149,150]. This underscores the importance of further research to find suitable oils for the formulation of oleogels in chocolates, a task that is significant in the food industry.

#### 2.3.2. Structuring Agents

The effective selection of proper structuring agents in forming oleogels cannot be overstated. These agents play critical roles in modifying oil droplet size and providing system stability [61,151,152,153]. They are further divided into two groups: low-molecular-weight agents (LMWGs) [154], which self-assemble to form a crystalline network and stable oil phase, and high-molecular-weight agents (HMOGs) [155], which are capable of forming three-dimensional networks through physical interactions, such as hydrogen bonds [156]. The proper selection of these agents is crucial due to their effectiveness in forming three-dimensional networks capable of trapping liquid oils and ensuring a stable gel [157,158].

### 2.4. Oleogels Crystallization

Structuring agents, crucial components dispersed in the oil phase, play a pivotal role in the formation of stable crystalline/polymeric networks [159,160]. Their reorganization during the cooling stage is a key factor. The formation of a stable gel is influenced by the nucleation and growth of crystals, which are sensitive to cooling rates. Higher rates result in denser and more uniform networks [161]. The concentrations of the structuring agents also play a significant role, allowing for the formation or loosening of crystalline structures and affecting their desired properties [162,163,164]. It has been demonstrated that the crystals formed in oleogels constitute a hierarchical structure similar to that of saturated fats [160].

For instance, as shown in Figure 3a, extracted from the report by Doan et al. [165], oleogels prepared from rice bran wax (RBW), sunflower wax (SW), beeswax (BW), candelilla wax (CLW), carnauba wax (CRWW) and berry wax (BEW) are the most effective, even at low concentrations (0.5% by weight). This is due to the presence of the β-polymorph, as detected via XRD, which is responsible for the formation of the stable crystalline network, as well as the β′ polymorph that results in nucleation, crystal growth, aggregation, and crystallization [164]. These findings not only enhance our understanding of oleogel formation but also open up new possibilities for the development of effective oleogels from various waxes, even at low concentrations, which can have significant implications in the food industries.

Another intriguing finding is that preparing oleogels using monoglycerides allows for more compact three-dimensional network gels. This novel strategy not only improves oleogel performance but also leads to antimicrobial solid properties, a significant advancement in the field [167,168]. Figure 3a presents other research on crystal networks in oleogel systems based on carnauba wax and monopalmitate, where more than one exothermic peak is attributed to the complex composition of the structuring agent [169]. Moreover, in oleogels prepared with glycerol monobehenate, small crystals in the form of needles and platelets (associated with stability and improved oleogel properties) were identified [167]. In addition, oleogels that use oil with a high content of polyunsaturated fatty acids, such as linoleic acid, lead to higher crystallinity and interfacial tension in terms of the crystal-melt, resulting in more stable structures [159]. Furthermore, in oleogels based on coconut oil, the addition of β-sitosterol increased the crystallinity, gel storage modulus, and hardness [163].

The properties of oleogels, particularly their flavor, texture, gloss, melting, and mouthfeel, are significantly influenced by the β-polymorph of cocoa butter [170]. This polymorph type, associated with 2-OH and 3-OH groups linked with hydrogen bonds [163,171], is what gives the oleogel its crucially stable crystalline network. Understanding and manipulating this factor is key to enhancing the quality of oleogels.

#### 2.4.1. Oleogel Crystallization Mechanism in Cocoa Butter

Figure 3b presents an innovative illustration from Li and Liu [166] that sheds light on the crystallization of CB mixed with an oleogel. As the figure illustrates, the combination of the oil phase and the structuring agent significantly raises the energy barrier for CB crystallization. This inhibits its nucleation and crystal growth, a finding supported by its isothermal crystallization kinetics. Notably, incorporating an oleogel into CB delays the crystalline transformation of β_V_ to β_VI_, thereby preventing fat bloom in chocolate and improving its stability during storage [169]. This practical application underscores the immediate relevance of this research to the food industry.

#### 2.4.2. Crystalline Morphology of CB and Other Oleogels

Figure 4 presents the microscopic analysis, unveiling the morphology of BC and three types of oleogels (P1). The oleogels’ oil phase is depicted as a dark zone, while the crystals stand out in bright white. The β-sitosterol with γ-oryzanol oleogel (A) exhibits a tubular structure that effectively traps the oil, β-sitosterol with stearic acid oleogel (B) forms a network of small crystals with smaller voids, and β-sitosterol with lecithin oleogel (C) presents dispersed crystals with large spaces, indicating lower stability. The CB (D) oleogel presents a spherical structure akin to that described by Li and Liu [34]. This analysis underscores the pivotal role of the structuring agent’s composition in dictating the oleogel assembly pattern, leading to the creation of diverse three-dimensional microstructures and opening up intriguing possibilities for their application.

Another characteristic observed through the use of polarized light microscopy is the change in the crystals on the surface (P2) of pure chocolate (S0) and with oleogels with different sugar concentrations (S1–S2). Crystal growth associated with fat bloom was reported, causing scattered white spots on the surface of the chocolate. A negative aspect of this is the appearance of a grayish hue due to scattered light [7]. In addition, the conventional chocolate presented more white spots than the chocolate samples containing oleogel. It should be noted that the use of oleogels containing palm sap sugar showed fewer white spots, inferring that this type of sugar in the oleogel limits the migration of the oil on the surface of the chocolate (bloom).

## 3. Contribution of Edible Oleogels to the Quality of Chocolates

### 3.1. Incorporation of Oleogels in the Production of Chocolates

According to Figure 5, to develop chocolates with the incorporation of oleogels, CB is first melted at 50 °C [24,27,34] or at 80 °C in a water bath [13]. Then, cocoa liquor, sugar, and other ingredients are added according to the type of chocolate being developed [172]. Then, it is homogenized using a ball mill for 2 to 2.5 h at 300 rpm [13,24,26,27] to achieve a particle size of approximately 20 μm [34]. After obtaining the chocolate base mixture, the oleogel is incorporated. According to Chen et al. [24], the oleogel is melted at 80 °C for a short time and then incorporated into the base mixture. In a similar manner, Li et al. [27] conducted a study where oleogels were melted at 8 °C and then mixed with the refined masses at the same temperature using magnetic stirring. Additionally, another study conducted by Li and Liu (2019) [34] indicated that after refining, the chocolate masses were mixed with melted oleogels (monoglyceride stearate and β-sitosterol and lecithin melted at 80 °C, ethylcellulose melted at 140 °C) on a magnetic stirrer within a few minutes. On the other hand, the study by Espert et al. [13] suggests incorporating oleogels by lowering the 80 °C temperature of the base mixture to 60 °C. On the other hand, Zhu et al. [26] incorporated oleogels into a base mix and refined it for 1.5 h.

Once the oleogel is incorporated into the chocolate mixture, tempering continues [13] through controlled cooling and reheating before pouring the mixture into molds. This process is crucial to obtain a glossy chocolate with a suitable texture [173]. Studies conducted by Chen et al. and Li et al. [24,27] suggest cooling the mixture to 32 °C and 28 °C and then reheating to 30 °C. Finally, before molding, air bubbles are removed using a vibrating table. At the end of the process, the molds are cooled (4 °C) and stored [27,34]. So far, studies indicate that chocolates made with oleogels have been stored in sealed bags and refrigerated for short periods [24,34,79]. However, it is clear that further research is needed to identify the optimal storage conditions, as factors such as variations in temperature and humidity can affect the stability, texture, and color of oleogel chocolate [174]. Inadequate temperature and humidity or constant fluctuations during storage can compromise the hardness of chocolate [175]. Therefore, the process of creating chocolates with oleogels shows promising results. However, the full understanding of the effects and benefits of these systems in chocolates is a key area that requires further research. Additionally, more studies are needed to understand and control the time and temperature parameters during the mixing, tempering, molding, and storage processes. This control is crucial to ensure the uniform distribution of the oleogel in the chocolate, prevent oil migration, and maintain product quality. Furthermore, environmental factors, such as light and oxygen, can significantly impact the texture, flavor, and stability of chocolate. Therefore, it is imperative to investigate the appropriate storage and packaging parameters of oleogel chocolates.

### 3.2. Oleogel Chocolates: A Review of Studies

In oleogel chocolates, the structuring agent not only shapes the oleogel’s structure but also has the potential to enhance the sensory characteristics of the final product [176], introducing flavors or aromas that consumers can savor [177]. However, the choice of structuring agent is not to be taken lightly. It is a critical decision that can significantly impact the homogenization of the mixture and the stability of the oleogel and chocolate [109].

#### 3.2.1. Structuring Agents and Oleogels Applied in Chocolates

This review has revealed that the most effective structuring agents for the formation of oleogels applied in chocolates are those corresponding to self-assembled structures of LMWGs (ii) and self-assembled structures of polymers or polymeric chains (iii) [13,34,39,178]. In turn, these structuring agents are preferred because of their ability to provide stability and maintain the desired texture in chocolate without significantly compromising its organoleptic properties [82,179,180]. However, some structuring agents that could fit into these categories still need to be sufficiently studied in the context of chocolate processing. Table 2 details the structuring agents used in oleogels with applications in chocolates. Table 3 reports the effect of the addition of oleogels in chocolates.

In the case of HPMC, this polymer is widely used as a structuring agent due to its solubility in polar organic solvents and its affordable cost [13,184,185,186]. HPMC is an amphiphilic polymer that facilitates the formation of oleogels, acting as a natural gelling agent that converts liquid oil into a solid gel through a cryogel template [184,186]. The process includes dissolving the HPMC in water, freezing to form the cryogel, freeze drying to sublimate the water, and mixing the cryogel with an oil. The porous structure of the cryogel, with numerous large pores, effectively captures the oil, favoring the production of oleogels [187]. Likewise, the use of HPMC is also an effective strategy to reduce the caloric content of foods [22]. Its use significantly improves texture and viscosity in oleogels that are suitable for chocolate production [13]. However, HPMC is pH-sensitive [188,189,190], decreasing its gelation capacity in acidic or alkaline environments. Moreover, its dispersion in the oil phase for gel formation is prolonged [191,192], suggesting that its application should be handled carefully.

Regarding beeswax, this animal-derived lipid containing fatty acids and long-chain alcohols [42] is widely used in the food industry. It stands out for its application as a structuring agent [176,182] due to its ability to structure oils [119]. In addition, waxes provide creamy textures and increased brightness [42,55,119] and preserve aromas for a long time by retaining volatile compounds [159].

β-sitosterol is a phytosterol that mimics the structure of cholesterol and needs to be mixed with other structuring gelling agents [39,193]. It is generally mixed with γ-oryzanol, lecithin, and monoglycerides, among others [27,34,39,193]. This combination is necessary because β-sitosterol alone does not form a network structure suitable for trapping the oil phase, as its aggregation and precipitation do not allow a stable network [194]. Despite this, β-sitosterol possesses various bioactive properties, such as anti-inflammatory, antioxidant, and antitumor effects [195]. However, it is essential to note that high concentrations of this structuring agent can affect chocolate flavor and aroma variations [196].

Ethylcellulose is a linear polysaccharide that is part of the readily available polymeric structuring used in the chocolate industry [59]. The characteristics that make it appreciated are its lack of color, odor, and taste [81]; thus, it does not affect the sensory attributes of chocolate. It also has high pH stability; however, its thermal stability under prolonged storage conditions is short [197,198]. The use of this structuring agent is usually in the form of combinations with other gelling and structuring agents, such as monoglyceric stearate and β-sitosterol + lecithin [27,34].

Finally, monoglyceric stearates are structuring agents that have multiple applications in food products, including chocolate [176,199], due to their ability to allow chocolate to melt and prevent fat migration [24]. These compounds are compatible with various oils, allowing uniform mixing and dispersion in the oleogel matrix [200] and providing benefits for adequate chocolate stability [201]. However, the structuring properties of monoglycerides can become unstable at extreme temperatures, either very high or very low, limiting their use in foods that require processing or storage under extreme conditions [202].

#### 3.2.2. Effect of CB Substitution with Oleogels on the Properties of Chocolates

Studies have focused on replacing CB with oleogels; specifically, Espert et al. [13] managed to replace 50% of CB with oleogel, reducing up to 39% of the saturated fatty acid content without altering the organoleptic properties of chocolates. Likewise, studies conducted by Chen et al. [24], Jin et al. [203], and Li et al. [27] showed that the use of oleogels allows for delaying fat bloom in chocolate due to the reduced number of surface crystals [204]. Therefore, the incorporation of oleogels as CB substitutes in chocolate production increased the hardness of the product [57]. Moreover, the presence of more polymorphic β-form [24] improved its rheological properties [57], gloss and color [26,39], soft melting properties, and shelf life [24].

On the other hand, the quality of chocolate is not only defined by its nutritional properties but also by its physicochemical, thermal, and sensory properties. Therefore, it is necessary to analyze them. In turn, it is necessary to determine the quality of an oleogel and its suitability for application in chocolate [180]. Table 4 reports the key attributes analyzed in relation to oleogels and oleogel-based chocolates.

As described in Table 4, the compatibility attributes of oleogels with chocolate are crucial [218,219] since they influence the organoleptic properties of the product [220]. Moreover, it is necessary to know the changes that occur in the chemical and physical composition after the incorporation of an oleogel in chocolate [166], such as crystal formation, miscibility, and stability at different temperatures [27,166,205], in order to identify the effects on its physical properties [26]. Tools such as rheology allow for the evaluation of the resistance to mechanical forces not only of an oleogel but also of chocolates containing oleogels [82]. On the other hand, it is necessary to identify the fatty acid composition of chocolates that contain oleogels [23] by using liquid and gas chromatography [181,206].

Another important property is the melting point, concerning the thermal behavior of a mixture of chocolate and an oleogel or an oleogel alone. Differential scanning calorimetry (DSC) measures the heat absorbed or released during the transition of states [207,219]. This measurement makes it possible to identify the optimal storage conditions for chocolate in order to retain its textural and thermal properties since oleogels increase chocolate’s melting point [27]. In addition, sensory analysis allows for the evaluation of organoleptic characteristics, such as the flavor, aroma, and texture of the final product [181].

In terms of mouthfeel and overall acceptability, a study conducted by Zhu et al. [26] showed that chocolates containing incorporated oleogel present higher values than those containing CB. Likewise, other studies report that chocolates with oleogels show an appearance similar to that of conventional chocolate but have significantly reduced saturated fatty acid contents [24,26,27,39], positioning oleogels as promising candidates for reducing the use of CB in chocolate manufacturing. This reduction in CB could be beneficial for human health, especially for frequent chocolate consumers. Another aspect is color (L*, a*, and b*, among others), which depends on the type and amount of structuring agent used [26]. Color is one of the key parameters used to evaluate the quality of chocolate at first sight [39,221]. Moreover, using oleogels as substitutes for CB improves the brightness of chocolates [39]. Chocolates containing oleogels achieve higher brightness (L*) scores than conventional chocolates [26]; however, the brightness of CB alone is superior to oleogels. In addition, the brightness of chocolates is related to fat bloom [27], which is mitigated by the use of oleogels [204]. Therefore, it can be concluded that chocolates containing oleogels will retain more of their brightness during storage compared to conventional chocolates. The color of chocolate is one of the key parameters used to evaluate the quality of chocolate and is perceived before flavor and aroma [221]. Therefore, further studies on the colorimetry of low-fat oleogel chocolate are essential as color can significantly influence the perceived sensory characteristics.

Figure 6 presents the results relating to chocolates containing oleogels extracted from studies conducted by Chen et al. and Espert et al. [13,24]. Regarding heat resistance (HR) after storage, the HR of conventional chocolate (A1) was between 25 and 30 °C, melting completely at body temperature (36 °C) [222]. On the other hand, using sugars and substituting 30% of CB with oleogels also increased the HR. Using sucrose (B1), maltitol (C1), tagatose (D1 and E1), and palm sap sugar (F1 and G1) yielded a melting HR of around 40 °C [24]. Regarding the visual appearance (VA), the appearance of conventional chocolate and chocolate containing oleogels at different HPMC concentrations (B2, C2, D2, and E2) was investigated, and it was reported that the chocolate presented a hard, cuttable, and homogeneous morphology, with a smooth and shiny external texture [13].

Likewise, Chen et al. [24] pointed out that chocolates containing added oleogel present in the β-form, which is the polymorphic form that gives chocolate its gloss, color, hardness, smooth melting, and shelf life characteristics. This suggests that an oleogel does not significantly affect the crystalline form of the chocolate. Moreover, the chocolates containing oleogel showed a lower intensity in the diffraction peaks because of the reduced CB and the different triacylglycerol (TAG) composition of the incorporated oleogels.

Regarding the cross-linking of the network, if an adequate three-dimensional structure is not established in the chocolate containing an oleogel [84], there may be oil leakage in the final product, affecting the structural properties of the food [223,224]. To evaluate this property, X-ray diffraction scattering can be performed [208] to avoid the leakage of oil [27] and chocolate rancidity. Through this analysis, it is possible to determine whether the method used for oleogel preparation was the most appropriate. Another aspect is the oil retention capacity, which is associated with the capacity of the oleogel to maintain the oil within its structure without separating [225]. High oil retention indicates good oleogel quality [226]. Likewise, this analysis allows for information on the oil binding capacity of the oleogel [27].

To gain insights into the phase distribution and organization of the oleogel components in both standalone and chocolate-incorporated forms, microstructural analysis is conducted using techniques such as optical/electron microscopy, image analysis for size and distribution assessments, X-ray diffraction for crystallinity evaluations, and in select cases, X-ray tomography for three-dimensional visualizations [227,228,229]. In chocolate manufacturing, these measurements are crucial to ensuring a homogeneous texture, optimal flavor release, and stability during storage.

It should be noted that Fourier transform infrared spectroscopy (FTIR) provides valuable insight into the structure and molecular interactions in composite materials [57]. In the case of oleogels, FTIR, when used in conjunction with X-ray diffraction (XRD) measurements, allows for a comprehensive characterization of oleogel structures [39]. This technique can be used to analyze the molecular interactions of the functional groups in oleogel systems by identifying the absorption peaks that correspond to specific functional groups [72]. This method is frequently employed to examine alterations in these peaks, offering insights into the interactions between molecules and functional groups present in oleogels [208].

It is crucial to measure the smoke point of an oleogel in the case of chocolate production. This method is used to determine the temperature at which the oil or fat begins to emit smoke continuously [217]. This parameter is crucial for ensuring the thermal stability of an oleogel, which is essential for maintaining the quality and flavor of the chocolate during critical processes, such as tempering and molding [214]. In conducting this analysis, a study conducted by Shahamati et al. [208] utilized the AOCS Method Cc 9a-48. This standardized method was developed by the American Oil Chemists’ Society (AOCS) for determining smoke, flash, and combustion points [230].

The analysis of chocolate containing oleogels allows us to understand and control the interactions between chocolate and oleogel components, ensuring quality control. As noted by Espert et al. [13] and Zhu et al. [26], using oleogels in place of CB in high quantities may impact the texture and flavor of chocolate. Therefore, it is essential to research the optimal ratio of CB to be substituted by the oleogel without compromising the stability of the chocolate product.

It is essential to note that since oleogels are made with various oils and gelling agents, people sensitive to specific components could experience allergic reactions. Therefore, in addition to knowing the specific types of oils and structuring agents used and how they interact to maintain the properties of chocolate, it is essential to understand the complete composition of the oleogel to prevent potential health risks. Given that the health properties of oleogels can vary and there are differences in regulations, products containing them must have clear and precise labeling, allowing consumers to identify possible risks and make informed decisions about their consumption.

### 3.3. Possible Hazards in Cusming in Chocolates with Oleogels

There is a common perception that fats have a negative impact on health, including the development of cardiovascular diseases and other pathologies, such as cancer, diabetes, and stroke [231,232]. However, despite this perception, triglycerides, composed of monounsaturated and saturated fatty acids, are an essential macronutrient and part of the human diet. In light of these considerations, there has been a notable increase in the prevalence of low-fat diets, which have been developed to replace trans fats with oleogels. It is important to note that oleogels, with their potential health benefits, are a promising alternative to trans fats.

In chocolate production, oleogels are an effective means of reducing saturated fatty acids and increasing the content of unsaturated fatty acids. This not only allows for greater control over cholesterol levels in consumers with cardiovascular pathologies [13] but also enhances nutrient absorption by boosting the oxidative stability of unsaturated oils [233]. Reducing the saturated fat content in chocolate, thanks to oleogels, may contribute to better regulation of carbohydrate metabolism, which would benefit people with type 2 diabetes [155,234]. Furthermore, the reduction in saturated fat can assist in regulating caloric intake. It is important to note that chocolate, even chocolate containing oleogels, remains a high-calorie food due to its sugar content. It is important to note that oleogels do not cure metabolic disorders. Instead, they represent a system that can improve the nutritional profile of chocolate [235].

While oleogels offer several advantages, the self-assembled structures within them can impede the interaction between lipase and lipids, which, in turn, slows fat metabolism and gastrointestinal digestion. The interaction is affected by the differences in crystal morphology and gel strength, which impact emulsification efficiency and the release rate of free fatty acids [236]. This could alter the intestinal microbiota and digestion. It is also important to note that oleogels may be more susceptible to lipid peroxidation due to their unsaturated fatty acid content and result in oxidative stress and inflammation, which are linked to the development of atherosclerosis, cancer, and inflammatory disorders, such as insulin resistance and arthritis [237]. CB has a more stable structure and contains natural antioxidants that protect it against oxidation.

An oleogel must exhibit controlled release properties within the digestive tract to effectively replicate the functions of fats. The release of lipid-soluble molecules from the oleogel is controlled by several factors inherent to oleogels. These factors include an oleogel’s texture and structure, which depend on the type and amount of structuring agents used in its preparation [219]. However, there is still no definitive answer to whether or not the integrity of the oleogel network can influence nutrient absorption in the gastrointestinal tract. Moreover, some of the structural agents utilized in oleogel production, such as waxes and hydrocolloids, may not be readily digestible by humans. The digestibility of oleogels is primarily concerned with the solvent’s impact on the gel’s mechanical disruption. Combining diverse structural agents and bioactive molecules can exert a synergistic or antagonistic effect [220]. Consequently, research should also prioritize the development of oleogels that are more compatible with human digestion.

The health benefits associated with oleogels are frequently attributed to their high concentration of unsaturated fatty acids compared to saturated fats. However, research on the safety of specific oleogels in cellular, animal, or human systems is limited [119,238]. To encourage the use of oleogels in the food industry, it is essential to guarantee the health benefits and safety of oleogel-containing foods under diverse processing conditions. However, it is equally important to be aware of the potential risks associated with oleogels, particularly the formation of hazardous substances during processing, which could impact the safety of the final product.

Consequently, the need for additional cytotoxicity studies and investigations in short- and long-term animal models to evaluate the safety and health benefits of foods incorporating oleogels is paramount. These studies are crucial to guaranteeing the safety of oleogels [21,239]. Therefore, further research must be conducted to understand the effects of oleogels on gastrointestinal health and metabolism. This will ensure that these products are appropriately evaluated and regulated to guarantee their safety and quality. Further bioassays and additional studies will provide more accurate data on the effects of oleogels, thus ensuring their safety and benefit to consumers.

## 4. Pectin as a Promising Source of Structuring Agent: A New Niche for Study

Considering health and food stability concerns, HMOGs, such as proteins and polysaccharides, have attracted scientific attention [28]. Their ability to form three-dimensional networks through hydrogen bonding means that oleogels possess viscoelastic properties, which are highly dependent on molecular weight, conformation, and polymer concentration [156]. Polysaccharides are composed of monosaccharides, such as glucose and fructose [240]. Two main strategies for oil structuring using polysaccharides can be found in the literature: direct or indirect [156]. Direct oil gelation using these polymers is not straightforward and involves the formation of three-dimensional networks of polysaccharides that trap and retain an oil in a matrix [241]. Therefore, oil gelation using polysaccharides is mainly achieved using indirect methods, such as emulsion templates, high internal phase emulsions, solvent exchange, drying, and freeze-drying [242]

However, chocolate researchers have yet to extensively study the oleo-gelling capacity of some polysaccharides, such as pectins. Pectins are obtained from the skins of fruits and vegetables, including citrus fruits [243,244,245]. These skins are agro-industrial by-products, with more than 10 million tons being generated annually after the production of juices, essential oils, and preserves [246,247]. This abundance facilitates access and utilization. The extraction of pectin from these by-products adds value to these materials and promotes their sustainable use, contributing to the mitigation of environmental problems [248]. In addition, pectins possess antioxidant properties [105]. However, incorporating citrus pectin oleogels could affect the flavor and aroma of chocolates [249].

Pectins are macromolecules mainly composed of polysaccharides, such as galacturonic acid, which is widely used as a gelling agent and a stabilizer of lipid structures [250,251,252,253,254,255]. They are also classified into high and low methoxyl pectin according to their degree of esterification [256,257,258], as detailed in Table 5.

The main challenge relating to its use is due to its hydrophilic nature and ability to form gels in aqueous environments [263,264]. However, a review conducted by Said et al. [265] points out that pectins present an interesting opportunity as structuring agents through the use of freeze-drying processes, homogenization methods, and the emulsion template method. This capability highlights pectins as a promising source of structuring agents in the food industry, opening a new niche of study for their application in oleogels in the production of chocolates.

### 4.1. Freeze-Drying

This procedure involves the initial formation of a stable pectin–oil emulsion. This product is frozen and then dehydrated via sublimation, generating a porous structure of pectin and oil that preserves the original emulsion’s structure and ensures the oleogel’s stability [266]. A study conducted by Luo et al. [105] on oleogels made with camellia oil, polyphenol–tea palmitate, and citrus pectin using the template emulsion method revealed that the higher the concentration of citrus pectin, the higher the density and hardness of the oleogel, thus improving its oil-holding capacity and gel strength.

On the other hand, Pan et al. [107] developed pectin oleogels combined with tea polyphenol esters with different fatty acid chain lengths and used them as fat replacers in cookies. The results showed that, when replacing butter with these oleogels, changes in the texture and sensory characteristics of the cookies were observed. However, in some instances, the cookies with oleogels presented qualities similar to those of traditional butter cookies, suggesting that these oleogels may be a viable alternative to replace fats.

### 4.2. Homogenization

The process of forming oleogels via homogenization involves mixing pectin with an oil phase to create an emulsion, where the pectin stabilizes the oil droplets in the aqueous phase. After homogenization, the emulsion is cooled to form the oleogel structure. This method is simple and scalable, allowing the gel structure to be adjusted by controlling the homogenization conditions. However, the resulting oleogels may have lower porosity and specific surface area compared to freeze-dried oleogels [265].

A study conducted by Liu et al. [41] points out that adding apple pectin can significantly improve the oxidative stability of oleogels because it forms a structural network that acts as a physical barrier against oxygen, thus reducing the rate of oxidation reactions. On the other hand, a study conducted by Dong et al. [266] showed that the addition of citrus pectin could make more stable oleogels, which occurs because citrus pectin improves the structure of the oleogel and strengthens the interface between the emulsion phases [253]. Moreover, studies show that pectin-based oleogels for cookies can reduce fat migration while maintaining an excellent sensory experience [107], offering an innovative strategy in terms of reducing saturated fats. However, there are no studies on pectin oleogels in chocolates, and information on the combination of pectin with other polysaccharides in the formation of oleogels is lacking [267].

A study conducted by Luo et al. [105] indicates that high methoxyl pectin shows great potential for oleogelification applications, being able to form stable emulsions over a wide pH range. Furthermore, according to Schmidt et al. [268], pectin with high (70% and 84%) and medium (55%) esterification degrees can form stable emulsions at pH ranging from 2 to 4. One of the sources of high methoxyl and high esterification degree pectin are citrus peels [269,270], such as orange, tangerine, and grapefruit [271,272,273], with 80.73% to 84.25% of galacturonic acid [274,275,276].

Although homogenization is more straightforward, the resulting oleogel may have a lower porosity and specific surface area than oleogels obtained via other methods, such as freeze-drying [265]. Therefore, further investigation of the optimal indirect methods for elaborating pectin-based oleogels, their bioactive contribution, and their applicability in chocolates is still required, adapting the method to the specific type of chocolate while considering both the properties of the chocolate and the oleogel. Therefore, there is a need for further research to understand their full impact on the sensory quality of chocolate. Thus, strategies must also be developed for adding additional compounds to adjust flavors and aromas [277] and improving consumer acceptance. In addition, it is essential to conduct further studies on the oleogenic capacity of the different citrus fruits present in each country and their application in the chocolate industry. Finally, exploring citrus diversity may also offer new opportunities in terms of optimizing the functionality of oleogels and improving the quality of chocolates, adapting them to the specific preferences and characteristics of each market.

## 5. Conclusions and Future Trends

Edible oleogels emerge as a promising alternative to saturated fats in chocolate production as they mimic the physical properties of CB while significantly reducing saturated fatty acid content. These gels, formed by an oily phase immobilized in a three-dimensional network of structuring agents, are crucial for ensuring sensory quality, product stability, and consumer health, making careful selection essential. Recent studies have shown that oleogels can prevent fat bloom, maintain smoothness and shine in chocolate, and offer a satisfactory sensory experience to consumers.

While oleogels made from waxes and polysaccharides such as EC have already proven effective as substitutes for saturated fats, the application of pectin-based oleogels in chocolates remains unexplored. Pectins, derived from citrus by-products, offer significant potential as structuring agents due to their self-assembly capability and oxidative stability. It is essential to explore various sources of pectin as structuring agents for oleogels and their impact on sensory quality, product acceptance, and lipid profile. Detailed studies are needed to assess the potential of citrus pectins in structuring oils and their use in chocolates, considering their role in reducing saturated fats and improving physicochemical, sensory, and thermal properties without compromising consumer health.

Although oleogels represent a viable and healthy alternative to saturated fats in chocolate production, formulations and processing conditions, especially those involving pectins, need optimization to maximize their potential and ensure high-quality products that meet consumer expectations and nutritional and health requirements. Comprehensive assessments of oleogel suitability for chocolate production are essential, including evaluations of structural stability, texture, fatty acid composition, and smoke point. Additionally, quality analysis must cover physicochemical, thermal, and sensory properties, compatibility with other ingredients, structural and rheological stability, and suitability for human consumption.

## Figures and Tables

**Figure 1 gels-10-00561-f001:**
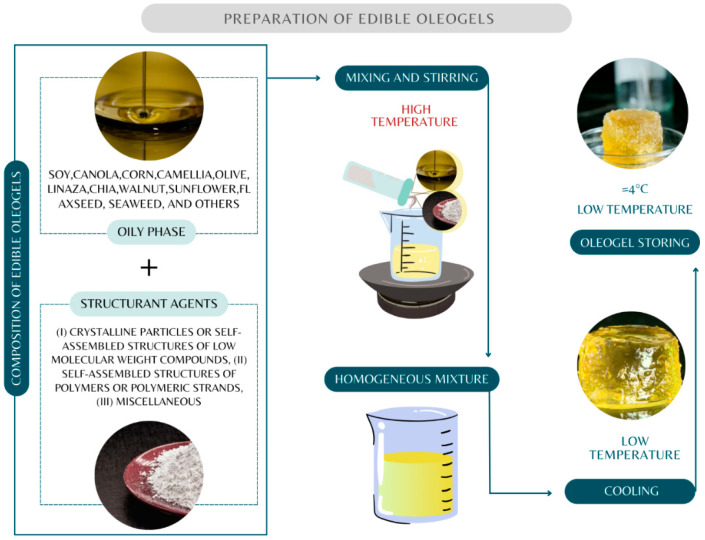
Description of edible oleogels.

**Figure 2 gels-10-00561-f002:**
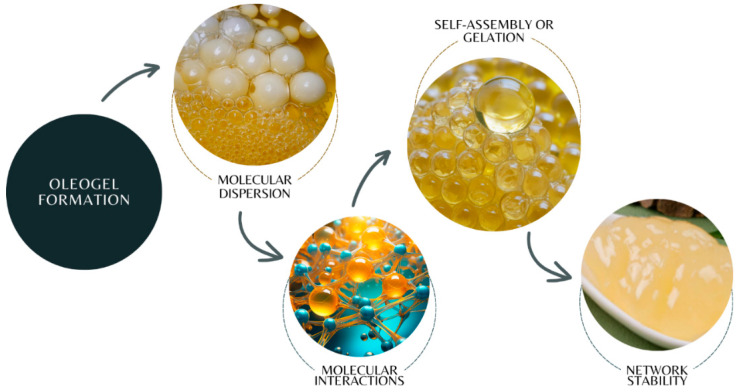
Stages of edible oleogel formation.

**Figure 3 gels-10-00561-f003:**
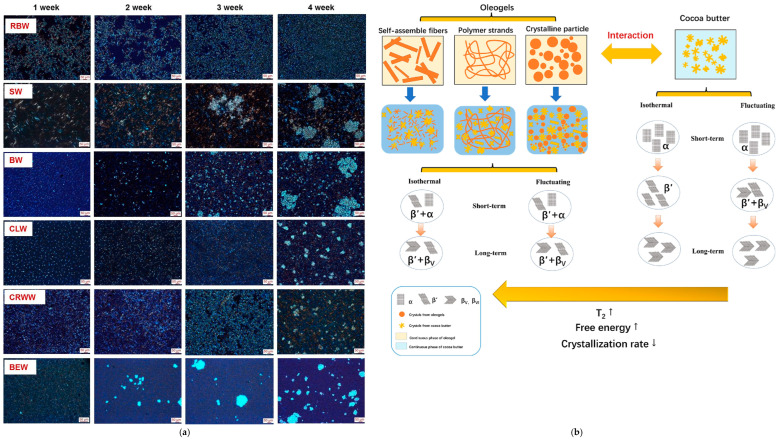
Crystalline morphology of oleogels (**a**) and mechanism of CB and oleogel crystallization (**b**) *. * Figures taken from Doan et al. and Li and Liu [165,166].

**Figure 4 gels-10-00561-f004:**
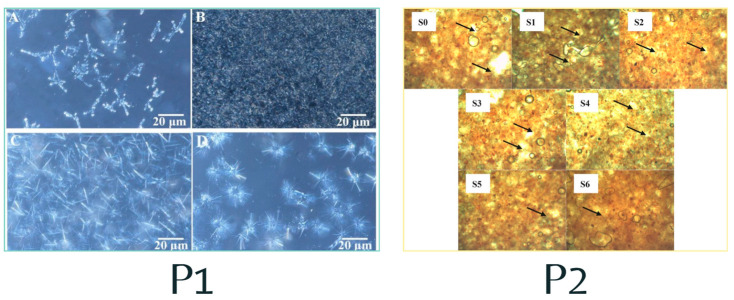
Microscopy of the crystalline morphology * of CB, oleogels, and chocolates with oleogels. * Image from Chen et al. and Sun et al. [24,39].

**Figure 5 gels-10-00561-f005:**
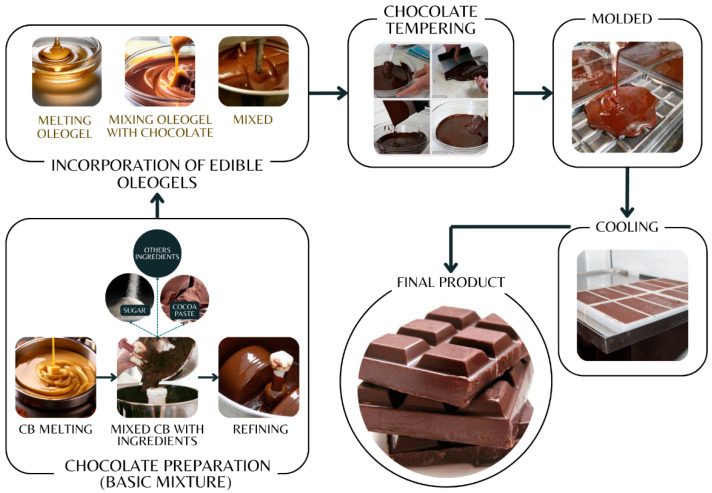
Chocolate manufacturing process with oleogel incorporation.

**Figure 6 gels-10-00561-f006:**
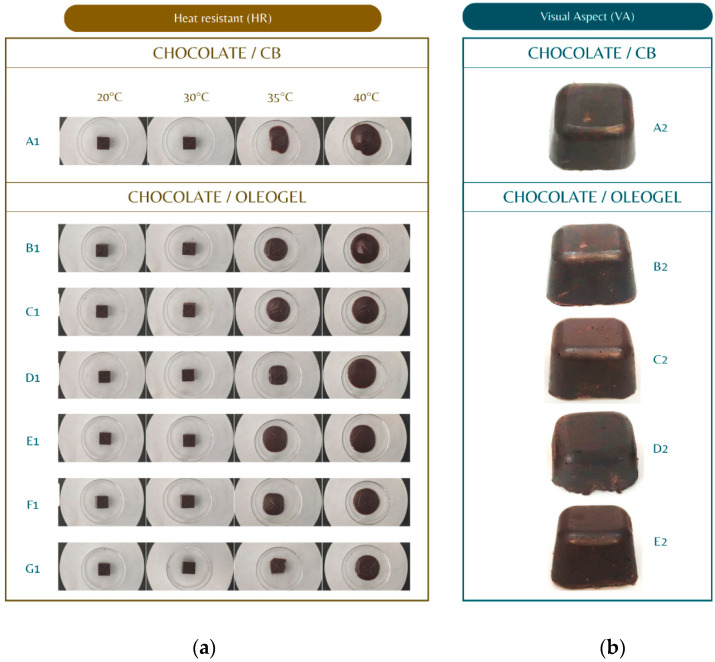
Heat resistance (**a**) and visual aspect (**b**) relating to oleogel chocolates *. * Figures taken from Chen et al. and Espert et al. [13,24]. HR: A1, pure chocolate (without oleogel); B1, chocolate containing oleogel and sucrose (100%); C1, chocolate containing oleogel and maltitol (100%); D1 and E1, chocolates containing oleogel and 5% and 10% tagatose, respectively; F1 and G1, chocolate containing oleogel and 25% and 50% palm sap sugar, respectively; VA: A2, control; B2, cocoa butter/oleogel chocolate (0.5% HPMC); C2, cocoa butter/oleogel chocolate (1% HPMC); D2 cocoa butter/oleogel chocolate (1.5% HPMC); E2 cocoa butter/oleogel chocolate (2% HPMC).

**Table 1 gels-10-00561-t001:** Composition of edible oleogels.

Oil Phase	Structuring Agents	Applications	References
Soy	Monostearin, sugar ester, candelilla wax, fully hydrogenated oils, monopalmitate, carnauba wax, ethylcellulose-2-highly unsaturated monoglycerides, beeswax, xanthan gum, pectin and gum Arabic, hydroxypropylmethylcellulose, glyceryl monostearate, monoacylglycerols fully hydrogenated oils, alginate, gelatin, and rice bran wax.	Baking cakes, pastries, hamburgers, and sausages	[79,85,86,87,88]
Canola (Colza)	Distilled monoglycerides, β sitosterol, γ oryzanol, ethylcellulose, hydroxypropylmethylcellulose, lecithin, candelilla wax, and carnauba wax–cellulose fibers–xanthan gum	Hamburgers, fried foods, baked goods, tortillas, cakes, cookies, and cakes	[71,89,90,91,92,93,94,95,96,97,98,99,100]
Corn	β sitosterol, ethylcellulose, oryzanol, stearic acid, lecithin-sodium stearyl lactate, polyglycerol esters, sorbitol monostearate, monoacylglycerol, carnauba wax, beeswax, and glycerol monostearate	Bakes, chocolate, cakes, cookies, and cold cuts	[34,39,85,101,102,103,104]
Camellia	Citrus pectin–beeswax.	Cookies, ice cream, and cakes	[52,105,106,107]
Sunflower	Diacylglycerol, triacylglycerol, lactylate, fruit wax, lecithin, behenic acid, stearic acid, beeswax, monoglycerides, ethylcellulose, methylcellulose, hydroxypropylmethylcellulose, monoacylglycerols and gum Arabic, stearic acid, glyceryl monostearate, polyglycerol stearate, β sitosterol, stearic acid, candelilla wax, egg white protein, ramnolipids, astaxanthin, glyceryl monostearate, and phytosterols.	Baked cakes, fried foods, pastries, chocolate, and chocolate spreads	[72,77,108,109,110,111,112,113,114,115,116,117,118,119]
Olive	Pig skin powder, monoglycerides, sunflower wax, and rice bran wax–candelilla wax–beeswax–phytosterols	Hamburgers, ice creams, cakes, buns, chocolate spreads, chocolate, and cookies	[38,120,121,122,123,124]
Chia	Monostearates–candelilla wax–beeswax	Sausages and bakes	[125,126,127]
Flaxseed	Sunflower wax and beeswax	Chicken meat biscuits, sausages, and cookies	[78,128,129]
Hazelnuts	Xanthan pectin–beeswax–monoglycerides	Hamburgers, bakery, and pastry	[85,130,131,132]
Coconut	Glyceryl monostearate	Fried foods	[133,134,135]
Sunflower–Olive	Soybean wax	Fried foods	[136]
Sunflower–Peanut–Corn–Flaxseed	Ethylcellulose	Sausages	[81]
Palm	Monoglyceride stearate	Chocolate, chocolate spreads, and fried foods	[24,137,138]
Hemp	Beeswax–rice bran wax–candelilla wax	Hamburgers and bakes	[139,140]

**Table 2 gels-10-00561-t002:** Structuring agents applied in the production of chocolates.

Structuring Agents	Characteristics					References
Hydroxypropylmethylcellulose (HPMC)	Solubility and dispersibility	Controllable viscosity	Thermal stability	Structuring capacity	Possible variation in flavor and aroma	[12,13,24,27,181,182,183]
Ethylcellulose (EC)	Maintain the appearance	Low viscosity		Structuring capacity	Possible flavor variation	
β-sitosterol	Heat resistance	Texture improvement	Thermal stability	Works with other structuring agents	Possible flavor variation	
Beeswax (BW)	Improves brightness	Viscosity increase		Texture limitations	Possible variation in flavor and aroma	
Monoglyceric stearate (MS)	Improves melting point	Texture improvement		Increases dispersion		

**Table 3 gels-10-00561-t003:** Oleogels (components and results) and their application in chocolates.

Oil Phase	Structuring Agents	Main Findings					References
High oleic sunflower oil	Hydroxypropylmethylcellulose	Decrease in hardness	Decreased melting point	Effect on sensory appearance	Saturated fats reduction	-	[13]
Palm oil	Monoglyceride stearate	Decrease in hardness	Better thermal stability	-	-	Fat bloom reduction	[24]
Corn oil	Monoglyceryl stearate, β-sitosterol + lecithin and EC	Increased hardness	Better thermal stability	-	Saturated fat reduction	Fat bloom reduction	[27]
Corn oil	Monoglyceryl stearate	Increased hardness	Good thermal stability	-	Saturated fat reduction	-	[34]
Pomegranate seed oil	Monoglycerides, beeswax, and propolis wax	Increased hardness	Low thermal stability	Good sensory appearance	Saturated fat reduction	-	[176]

**Table 4 gels-10-00561-t004:** Key attributes to determine the quality of chocolates containing oleogels.

Attributes	Evaluate	Analysis	References
Compatibility of oleogel and chocolate	Sensory acceptance and food safety of oleogel/chocolate	Chemical composition analysis and sensory analysis	[13,23,24,181]
Structural stability, texture, and consistency	Shape and structure of oleogel/chocolate	Rheology, liquid and gas chromatography	[57,82,180,181,205,206]
Melting point	Composition of fatty acids of oleogel/chocolate	Differential scanning calorimetry (DSC) gas chromatography	[27,166,180,181,207,208,209,210,211,212,213,214,215,216,217,218,219]
Network reticulation	The temperature at which solid oleogel/chocolate begins to melt	Nanoscopy	[210,211]
Oil-holding capacity	Strength and stability of the oleogel structure that traps oil	Centrifugation Soxhlet and Goldfish	[105,212,213,214,215,216]
Interactions and molecular structure	Molecular interactions and structural information of the oleogel	FTIR (Fourier transform infrared spectroscopy)	[39,168,208,212]
Smoke point	Thermal stability and quality of the oil/oleogel	[AOCS Method Cc 9a-48]	[134,208,217]

**Table 5 gels-10-00561-t005:** Main characteristics of pectin.

	Composition	References
Monosaccharides	ᴅ-galactose, ʟ-ramnose and ʟ-arabinose	[259]
Polysaccharides	Homogalacturonan and rhamnogalacturonan	[260]
Low methoxyl pectin	Less than 50% esterification degree	[255,261]
High methoxyl pectin	More than 50% of esterification degree	
Applications	Gelling agent, thickener, stabilizer, emulsifier, prebiotic, dietary fiber in many foods, and oleogelant	[105]
Gelling capacity	It varies according to the stages of growth, maturity levels of the plant of origin and fruit.	[262]

## Data Availability

No new data were created or analyzed in this study. Data sharing is not applicable to this article.

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
