# Peer review of "Oleogel Systems for Chocolate Production: A Systematic Review"

_gels, 2024, doi:10.3390/gels10090561_

Round 1

Reviewer 1 Report

Comments and Suggestions for Authors

Dear Authors:

I had the opportunity to read your review titled “Advances in oleogel systems for chocolate production: an up-date review.” Sorry to say but, from my perspective, the information discussed in the manuscript does not fulfill the in-deep scientific/technological discussion required to be a serious review about the use of oleogels in chocolate production. The manuscript present just a collection of small pieces of information in several topics, but the essence of the use of oleogels in chocolate is just superficially discussed, and sometimes erroneously (i.e., pages 2, lines 90-92; page 3, lines 106-113; page 6, lines 219-222) or simplistically discussed. The introduction section is a clear example for this, but additional examples are vastly distributed throughout the manuscript. The manuscript has 252 references, many without any relationship with the subject mater of the review or out of the context of the discussion. It is also unfortunately, that the manuscript present mixed/wrong information about the use of pectins to develop oleogels. Within this context, I cannot recommend the acceptance of the review. My best advice is that the authors make a thorough reading about the development and tentative use of oleogels in edible products, and also about the chemistry and science behind the technology of chocolate production. From there you could write a in-deep review about the exciting use of oleogels in chocolate products. 

Comments on the Quality of English Language

The use of the English language needs a revision throughout the manuscript.

Author Response

Dear Reviewer, we have considered all the suggestions that were made for you. Thank you for your support, below is a detailed response to each suggestion.

I had the opportunity to read your review titled “Advances in oleogel systems for chocolate production: an up-date review.” Sorry to say but, from my perspective, the information discussed in the manuscript does not fulfill the in-deep scientific/technological discussion required to be a serious review about the use of oleogels in chocolate production. The manuscript presents just a collection of small pieces of information in several topics, but the essence of the use of oleogels in chocolate is just superficially discussed, and sometimes erroneously (i.e., pages 2, lines 90-92; page 3, lines 106-113; page 6, lines 219-222) or simplistically discussed. The introduction section is a clear example for this, but additional examples are vastly distributed throughout the manuscript.

Response: Thank for the comment. In the main text we added discussion scientific/technological about the use of oleogels in chocolate production, view in text with red color [modifications]. The introduction was modified [L. 25 – 91], and we rewrite the information in the lines: L. 93 – 99, L. 107 – 124, L. 152 – 179]

The manuscript has 252 references, many without any relationship with the subject matter of the review or out of the context of the discussion.

Response: Thank you for your suggestion, we were worked in selected the references with relationship with the subject matter of the review or out of the context of the discussion, view in text with red color [modifications].

It is also unfortunately, that the manuscript present mixed/wrong information about the use of pectins to develop oleogels.

Response: Thank you for your comments, we were worked in improvements about the use of pectins to develop oleogels [L. 528 – 615]

Within this context, I cannot recommend the acceptance of the review. My best advice is that the authors make a thorough reading about the development and tentative use of oleogels in edible products, and also about the chemistry and science behind the technology of chocolate production. From there you could write a in-deep review about the exciting use of oleogels in chocolate products.

Response: However, we have worked to improve the manuscript based on your comments. we added information included figures about the development and tentative use of oleogels in edible products, and also about the chemistry and science behind the technology of chocolate production [L. 191 -199, L. 216 – 220, L. 232 – 255, L. 258 – 289, L. 298 – 343, L. 345 – 469]

Reviewer 2 Report

Comments and Suggestions for Authors The prposed review provides a comprehensive overview of the use of oleogels as a substitute for cocoa butter in chocolate production. The authors convincingly presented all the advantages of using oleogels instead of coca butter: One of the primary motivations for using oleogels in chocolates is their potential to reduce saturated fat content. Cocoa butter, which is high in saturated fats, contributes significantly to the calorie content of chocolate (approximately 500 kcal/100 g)​​. By replacing coca butter with oleogels, the saturated fatty acid content in chocolates can be reduced significantly, which is beneficial for consumers' cardiovascular health. The reduction in saturated fats directly correlates with a decrease in health risks associated with high cholesterol, cardiovascular diseases, obesity, and possibly cancer​​. Therefore, oleogel-based chocolates could be a healthier alternative, particularly for individuals with cardiovascular concern. Research presented by the authors of the review has shown that consumers often cannot distinguish between conventional chocolates and those made with oleogels in terms of mouthfeel and appearance​. This suggests that oleogels can replace cocoa butter without compromising the sensory qualities that consumers enjoy.   Unfortunately, the authors presented only the positive sides of oleogels in their review. Therefore, in my opinion, the text of the article is biased and is not suitable for publication in its current form. The authors should also present, based on the literature on the subject, each possible health (and social) risks associated with the use of oleogels. Here you have a list of subjects, which, in my opinion, should be also describe/discussed in this paper: - The body's ability to absorb and utilize the nutrients in oleogels might differ from that of cocoa butter, potentially affecting the nutritional benefits. - Some studies suggest that oleogels could affect gut health differently. The digestion and absorption processes of oleogels could influence gut microbiota and gastrointestinal health​  - Replacing cocoa butter with oleogels might impact the metabolism of fats differently. This could have implications for individuals with metabolic disorders such as diabetes​ - Oleogels are often made from various oils and gelling agents, which could introduce new allergens into the diet. Individuals sensitive to specific plant oils or additives used in oleogels might experience allergic reactions​  - Oleogels may be more prone to oxidation compared to cocoa butter, potentially leading to rancidity and the formation of harmful oxidation products if not properly stabilized​. - The different structural and compositional properties of oleogels could affect their susceptibility to microbial contamination and growth, posing a potential food safety risk​
- The introduction of oleogels in chocolates requires careful consideration of regulatory standards and labelling requirements. Misleading health claims could lead to consumer mistrust​ ​.                  

Author Response

Dear Reviewer, we have considered all the suggestions that were made for you. Thank you for your support, below is a detailed response to each suggestion.

The proposed review provides a comprehensive overview of the use of oleogels as a substitute for cocoa butter in chocolate production. The authors convincingly presented all the advantages of using oleogels instead of coca butter: One of the primary motivations for using oleogels in chocolates is their potential to reduce saturated fat content. Cocoa butter, which is high in saturated fats, contributes significantly to the calorie content of chocolate (approximately 500 kcal/100 g)​​. By replacing coca butter with oleogels, the saturated fatty acid content in chocolates can be reduced significantly, which is beneficial for consumers' cardiovascular health. The reduction in saturated fats directly correlates with a decrease in health risks associated with high cholesterol, cardiovascular diseases, obesity, and possibly cancer​​. Therefore, oleogel-based chocolates could be a healthier alternative, particularly for individuals with cardiovascular concern.

Research presented by the authors of the review has shown that consumers often cannot distinguish between conventional chocolates and those made with oleogels in terms of mouthfeel and appearance​. This suggests that oleogels can replace cocoa butter without compromising the sensory qualities that consumers enjoy.   Unfortunately, the authors presented only the positive sides of oleogels in their review. Therefore, in my opinion, the text of the article is biased and is not suitable for publication in its current form. The authors should also present, based on the literature on the subject, each possible health (and social) risks associated with the use of oleogels.

Response: Thank you for the comment.

Here you have a list of subjects, which, in my opinion, should be also describe/discussed in this paper:

  • The body's ability to absorb and utilize the nutrients in oleogels might differ from that of cocoa butter, potentially affecting the nutritional benefits.

Response: Thank for the comment. We added information [L.500 - 511]

  • Some studies suggest that oleogels could affect gut health differently. The digestion and absorption processes of oleogels could influence gut microbiota and gastrointestinal health​.

Response: Thank. We added information in the lines 490 – 499].

  • Replacing cocoa butter with oleogels might impact the metabolism of fats differently. This could have implications for individuals with metabolic disorders such as diabetes.

Response: Thank you for the suggestion. We added information [L. 484 – 489]

  • Oleogels are often made from various oils and gelling agents, which could introduce new allergens into the diet. Individuals sensitive to specific plant oils or additives used in oleogels might experience allergic reactions​

Response: Thank. We added information [L. 462 – 527]

  • Oleogels may be more prone to oxidation compared to cocoa butter, potentially leading to rancidity and the formation of harmful oxidation products if not properly stabilized​

Response: Thank for the comment, we added information [L. 494 - 499]

  • The different structural and compositional properties of oleogels could affect their susceptibility to microbial contamination and growth, posing a potential food safety risk

Response: Thank for the suggestion, we added the information [L. 512 – 527]

  • The introduction of oleogels in chocolates requires careful consideration of regulatory standards and labelling requirements. Misleading health claims could lead to consumer mistrust.

Response: Thank you for the comment, we added information [L. 46 – 50, L. 101 - 104]  

Reviewer 3 Report

Comments and Suggestions for Authors

The Authors presented an overview of possible applications of the oleogel system for chocolate production. The manuscript is based on 252 current references which is beneficial. I really enjoyed the topic of this overview, however major revision is needed.

First of all, there is no image/figure of how the chocolate with oleogels looks like - this must be presented.

Secondly, there is a lack of information on how the chocolate with oleogels compositions was made (i.e. some recipes and some visualizations step by step).

Please include at the beginning of the paper the "road map" of your paper (i.e. as a figure) what are the major points presented and described here - it should be more eye-catching.

More figures are desirable in this manuscript content.

Please add some comments on how the oleogels can impact the color of the final chocolate products.

 Please add some comments about the chocolate desirable properties as well.

Author Response

Dear Reviewer, we have considered all the suggestions that were made for you. Thank you for your support, below is a detailed response to each suggestion.

The Authors presented an overview of possible applications of the oleogel system for chocolate production. The manuscript is based on 252 current references which is beneficial. I really enjoyed the topic of this overview, however major revision is needed.

Response: Thank you for the suggestions.

  • First of all, there is no image/figure of how the chocolate with oleogels looks like - this must be presented.

Response: Thank for the suggestion. We added the figure [Figure 4, Figure 5, and Figure 6]

  • Secondly, there is a lack of information on how the chocolate with oleogels compositions was made (i.e. some recipes and some visualizations step by step).

Response: Thank for the comment. We added the information and figures about how the chocolate with oleogels compositions was made [L. 258-289, Figure 5]

  • Please include at the beginning of the paper the "road map" of your paper (i.e. as a figure) what are the major points presented and described here - it should be more eye-catching.

Response: Thank for the suggestion. We changed the graphical abstract according to manuscript information.

  • More figures are desirable in this manuscript content.

Response: Thank you for suggestion. We added more figures [Figure 3, Figure 4, Figure 5, Figure 6]

  • Please add some comments on how the oleogels can impact the color of the final chocolate products.

Response: Thank you for recommendation, we added information about impact the color of the final chocolate products [L. 384 - 395]

  • Please add some comments about the chocolate desirable properties as well.

Response: Thank you for recommendation, we added comments about the chocolate desirable properties as well [L. 298 – 343, L. 345 – 469]

Reviewer 4 Report

Comments and Suggestions for Authors

The manuscript is very interesting to scientists working on food industry and  to those who are going to get involved with oleogels. To my opinion it can be published after minor revision

Minor comments:

1.      FTIR is a common technique used to provide structural information about molecular interactions in oleogel systems. Incorporate this technique into the text and describe some examples.

2.      Another important subject is the polymorphism of oleogel samples. Was this observed in oleogel system for chocolate production? If yes, give some examples and comment the XRD results

3.      Discuss the color characteristics oleogel system for chocolate production

4.      Provide microscopy images showing the crystal morphology of CB and other oleogels

Other Comments:

Line 269: “Hidroxipropilmetilcellulose”  replace with “Hydroxypropylmethylcellulose”

Line 276: “Etilcellulose” replace with  “Ethylcellulose”

Author Response

Dear Reviewer, we have considered all the suggestions that were made for you. Thank you for your support, below is a detailed response to each suggestion.

The manuscript is very interesting to scientists working on food industry and to those who are going to get involved with oleogels. To my opinion it can be published after minor revision.

Response: Thanks you for the comments

Minor comments:

  1. FTIR is a common technique used to provide structural information about molecular interactions in oleogel systems. Incorporate this technique into the text and describe some examples.

Response: Thank you for recommendation, we incorporated this technique [L. 358 in Table 4, L. 439 - 447]

  1. Another important subject is the polymorphism of oleogel samples. Was this observed in oleogel system for chocolate production? If yes, give some examples and comment the XRD results

Response: Thank you for the comment, we added information about polymorphism of oleogel samples and chocolate with oleogel [L. 191 -199, L. 216 – 218, L. 351 – 354, and Figure 3]

  1. Discuss the color characteristics oleogel system for chocolate production

Response: Thank you for recommendation, we added discussion about the color characteristics oleogel system for chocolate production [L. 384 - 395]

  1. Provide microscopy images showing the crystal morphology of CB and other oleogels

Response: Thank you for suggestion. We added microscopy images showing the crystal morphology [L. 232 – 255, and Figure 4]

Other Comments:

  • Line 269: “Hidroxipropilmetilcellulose” replace with “Hydroxypropylmethylcellulose”

Response: Thanks, sorry for the mistake. We changed for “hydroxypropylmethylcellulose” [L. 155 in Table 1, L. 308 in Table 2, L. 309 in Table 3]

  • Line 276: “Etilcellulose” replace with “Ethylcellulose”

Response: Thanks, sorry for the mistake. We changed for “ethylcellulose” [L. 132, L. 330]

Round 2

Reviewer 2 Report

Comments and Suggestions for Authors

no additional comments

Author Response

Thank you for your comments. We have worked as recommended and addressed minor revisions from the editor.

Reviewer 3 Report

Comments and Suggestions for Authors

Dear Authors

The manuscript content was updated due to my and other Reviewer's comments. In my opinion, it is more eye-catching, and a lot of interesting information which is desirable in food technology, now are visualized (well done).

Overall my final recommendation is accept.

Additional remark and question (which can be dealt with at the editor stage) - did you perform the figures by yourself or taken from other work? If taken please add references to each figure.

Author Response

The manuscript content was updated due to my and other Reviewer's comments. In my opinion, it is more eye-catching, and a lot of interesting information which is desirable in food technology, now are visualized (well done).

Response: Thank yoy for the comment.

Overall my final recommendation is accept.
Response: Thank you for the decision 

Additional remark and question (which can be dealt with at the editor stage) - did you perform the figures by yourself or taken from other work? If taken please add references to each figure.
Response: Thank you for the suggestion. We added the reeferences in the main text (L. 227, 272 and 446), also we added copyrigths (see to atach files)
